# Current Pathology Model of Pancreatic Cancer

**DOI:** 10.3390/cancers14092321

**Published:** 2022-05-07

**Authors:** Krzysztof Szymoński, Katarzyna Milian-Ciesielska, Ewelina Lipiec, Dariusz Adamek

**Affiliations:** 1Department of Pathomorphology, Jagiellonian University Medical College, 31-531 Cracow, Poland; dariusz.adamek@uj.edu.pl; 2Department of Pathomorphology, University Hospital, 30-688 Cracow, Poland; katarzyna.milian-ciesielska@uj.edu.pl; 3M. Smoluchowski Institute of Physics, Jagiellonian University, 30-348 Cracow, Poland; ewelina.lipiec@uj.edu.pl

**Keywords:** pancreatic adenocarcinoma, pathology reporting, pancreatic neural remodeling, pancreatic cancer heterogeneity, morphological subtyping, pancreatic cancer spectroscopy

## Abstract

**Simple Summary:**

Pancreatic cancer is a very aggressive and lethal malignant neoplasm with overall 5-year survival rates below 10%. The field of pancreatic cancer research is rapidly evolving. Reports of newly revealed pathomechanisms of the nature of these tumors are published daily. Nevertheless, many aspects of a pathologic evaluation are still uncertain. It is crucial to be able to pull out practical information that impacts the diagnostic process, called a pathologic evaluation. In this review, we comprehensively summarize some of the recent papers from the pathologists’ and clinicians’ points of view. We specifically focus on pathology assessment and reporting, to make them meaningful for clinical and research purposes. Lastly, we highlight novel diagnostic and research approaches, point out some missing pieces in the field, and suggest further study directions.

**Abstract:**

Pancreatic cancer (PC) is one of the most aggressive and lethal malignant neoplasms, ranking in seventh place in the world in terms of the incidence of death, with overall 5-year survival rates still below 10%. The knowledge about PC pathomechanisms is rapidly expanding. Daily reports reveal new aspects of tumor biology, including its molecular and morphological heterogeneity, explain complicated “cross-talk” that happens between the cancer cells and tumor stroma, or the nature of the PC-associated neural remodeling (PANR). Staying up-to-date is hard and crucial at the same time. In this review, we are focusing on a comprehensive summary of PC aspects that are important in pathologic reporting, impact patients’ outcomes, and bring meaningful information for clinicians. Finally, we show promising new trends in diagnostic technologies that might bring a difference in PC early diagnosis.

## 1. Introduction

Pancreatic cancer is one of the most aggressive and lethal malignant neoplasms, ranking in seventh place in the world in terms of the incidence of death [1] and it is projected to surpass breast, prostate, and colorectal cancers to become the second leading cause of cancer-related deaths by 2030 [2]. It was estimated that there were 466,003 new pancreatic-cancer-related deaths in 2020 worldwide [3]. Significant improvements in diagnosis and management have not improved 5-year survival rates, which remain below 10%. The reason for this fact, among others, is the tumor heterogeneity including molecular aberrations, but also the tumor nature, appearing as a wide spectrum of patterns involving cancer gland formation and tumor stroma composition, which is much underrated among pathologists. The knowledge about processes and interactions between cancer cells and the surrounding tumor stroma microenvironment is rapidly expanding. It is of crucial importance that pathologists follow this trend and include the newest discoveries in their daily workup (Figure 1).

This review aims at summarizing the current knowledge concerning pancreatic ductal adenocarcinoma (PDAC), which is the most common form of cancer in the pancreas. Many publications describe in detail the pathomechanisms of this complex entity [4,5,6,7,8,9,10,11,12,13,14,15], but none of them focus on pathologic evaluation comprehensively. We attempt to summarize some of the recent papers from the pathologists’ and clinicians’ points of view. We will specifically focus on pathology assessment and reporting as this field, in our opinion, has many inconsistencies.

## 2. Contents Summary

This text is divided into sections. In the first two parts (Section 3 and Section 4), we briefly outline the pathomechanisms of PDAC development via the process described as “pancreatic intraepithelial neoplasia” (PanIN) and “intraductal pancreatic mucinous neoplasm” (IPMN) and show already proposed molecular subtype classifications. Following (Section 5), the importance of proper histopathologic evaluation is shown, starting with the description of the standardized examination protocol (LEEPP—Leeds Pathology Protocol).

Pancreatic cancer morphological heterogeneity is a well-known fact among pathologists. Although, the World Health Organization classification of tumors of the digestive system from 2019 [16], groups pancreatic carcinoma into conventional variants, it leaves over 90% of them entitled as “not otherwise specified (NOS)”. The next section (Section 6) describes some important aspects regarding the PDAC variants distinguished by the WHO classification and others described elsewhere. Specifically, we focus on subtypes of IPMNs, IPMN-mimickers, differentiation of “concomitant to” and “derived from” IPMN invasive tumors, proposed morphological subtyping by some authors [17,18], and highlighting other variants, such as clear cell, foamy gland, or large duct variants.

Subsequently (Section 7) we present to the reader some novel methods in immunohistochemistry (IHC) of PDAC, outlining the cellular malignancy markers (Maspin, IMP3, S100P, p53, loss of pVHL), specific subtype markers (HNF1B, CDX2), or the ones that stain the stromal compartment (LIF and IL-6).

Next, we focus on PDAC stroma, which continuously “cross-talk” with the pancreatic cancer cells (Section 8). In this context, cancer-associated fibroblasts (CAFs) and cancer stem cells (CSCs) are described, including their heterogeneity and impact on tumor malignancy potential.

In Section 9 and Section 10, we discuss a pathologic aspect of PDAC prognosis, which varies with morphological subtypes and immunohistochemical marker expression. Multiple factors influence the prognosis of PDAC patients that underwent curative surgery. Recently, the International Collaboration on Cancer Reporting (ICCR) published consensus guidelines, revealing so-called core and non-core elements of pathology reporting [19] that worsen the prognosis. In these sections, we briefly elaborate on some important aspects summarized in the ICCR dataset, as well as some others not included in that document.

Section 11, highlights different sides of perineural invasion (PNI), a topic that is being widely investigated in the PC research field. Further (Section 12), the problem of direct vs. metastatic lymph node involvement is briefly presented.

The final parts (Section 13 and Section 14) are a short glance into the newest PDAC diagnostic options. The interesting question arises as to whether super-precise serum biomarker detection technologies help in PDAC early diagnosis?

## 3. Precursor Lesions

Pancreatic adenocarcinoma may arise in several precursor lesions, that tend to have distinct molecular bases and are characterized by morphologically divergent entities. The most common are pancreatic intraepithelial neoplasias (PanINs) and intraductal papillary mucinous neoplasms (IPMNs).

PanIN, which is a flat (non-tumoral) form of dysplasia, develops in the ductal epithelium localized in “normal” pancreatic tissue, with or without signs of chronic pancreatitis or with so-called “acinar-to-ductal metaplasia” (ADM) regions [20,21,22,23,24]. It is considered a main preinvasive lesion for PDAC. The road from normal ductal epithelium to PDAC through PanIN requires sequential genetic alterations initiated by the KRAS mutation, which takes place in the earliest stage of the process [24,25,26]. Dysplastic changes progress from low-grade PanIN to high-grade PanIN and eventually invasive PDAC by a sequence of further inactivation of tumor-suppressor genes such as TP53, transforming growth factor-beta (TGF-β) signaling genes (SMAD4, TGFBR1, TGFBR2) or cyclin-dependent kinase inhibitor 2A (CDKN2a) [27,28,29]. Other described mutations include chromatin regulators (ARID1A) [30]. Additionally, the quiescent (normally suppressed in adults) Notch signaling pathway is being activated [31].

IPMNs are mucinous cystic tumors that arise from pancreatic ducts (main or branch ducts) and often cause symptoms due to duct obstruction (especially the main duct IPMN). A great role in IPMN development plays a guanine nucleotide-binding protein, alpha stimulating activity polypeptide (GNAS) proto-oncogene mutation, which is not found anywhere else in pancreatic tumors [32,33,34,35,36]. Briefly, GNAS mutations decrease the GTPase activity of the α-stimulatory subunit of the G protein (Gsα). In consequence, the cyclic adenosine monophosphate (cAMP) is constantly stimulated and via multiple effectors activates downstream signaling including the protein kinase A (PKA), exchange proteins directly activated by cAMP (EPAC1, EPAC2), or cyclic nucleotide-gated ion channel (CNG) pathways [37,38,39,40]. It was shown in a mouse model that constant PKA activation mediates the inhibition of salt-inducible kinases (SIK), important tumor suppressors. Thus GNAS mutations are critical in sustaining tumorigenesis in IPMNs [41].

There are three subtypes of IPMNs, divided mainly by their morphology: gastric-type, intestinal, and pancreato-biliary. GNAS mutations are found in every subtype of IPMN, but particularly frequently in intestinal-type IPMNs (75%) [42] and in the invasive colloid carcinoma that arises from an intestinal-type IPMN (83–89%) [35,43]. Interestingly, these lesions appear to have a better prognosis than non-colloid PDACs (see Section 9).

In the progression from IPMN into the “IPMN derived from” invasive carcinoma (IPMC), further molecular alterations are found, including early KRAS mutations (in approximately 65% of gastric-type IPMNs) [42]. It is suggested that these KRAS and GNAS mutations play a critical role in separating the evolution paths leading to pancreatobiliary-type IPMN and further into tubular invasive carcinoma, or intestinal-type IPMN into colloid invasive carcinoma [10].

Another event in IPMN progression includes RNF45 inactivating mutations that lead to enhanced Wnt/beta-catenin signaling and might play a role in gastric-type to intestinal-type IPMN progression (low-grade to high-grade progression) [42]. Recently Fujikura et al. identified KLF4 as a new driver gene in the IPMN progression model observed much more frequently in low-grade lesions [44].

Some authors suggest that gastric-type IPMN is a low-grade precursor form of the other two: intestinal-type and pancreatobiliary-type [45,46]. Omori et al. [47] showed the emergence of intestinal features directly in gastric-type IPMNs in 80% of studied cases and described evolution pathways of gastric-to-intestinal epithelium, that were driven by caudal type homeobox 2 (CDX2). The expression of CDX2 preceded the intestinal features and mucin 2 (MUC2) expression [47].

Many cases reveal multifocal IPMN tumors, as much as invasive carcinomas, occurring as a collision PDAC, growing independently from the IPMN (concomitant) or the one that grows out of the IPMN tumor (derived from). Divergent models of evolution for pancreatic dysplastic lesions were confirmed by Omori et al. [48], who identified three different progression pathways from IPMN to PDAC depending on molecular aberrations analysis. The first one included the “sequential” subtype with the same driver mutations among PDAC and concurrent IPMN, including GNAS mutations. “Branch-off” subtypes consisted of the same KRAS mutations but different GNAS mutations. The third called “de novo” had mutations found in PDACs but not found in concurrent IPMNs, which suggested different early clones.

An oncogenic KRAS mutation is present in over 90% of conventional PDACs (cPDAC) and in about 65% of IPMCs (with 80–90% in tubular IPMCs, and 30–50% in colloid IPMCs) [30]. It is an early but critical event in pancreatic carcinogenesis, required for initiation of the process, its maintenance, and progression. Consequently, there is a constitute activation of RAS downstream signaling pathways, such as the MAPK and PI3K-mTOR pathways. This deregulates key cellular processes that lead to uncontrolled cell proliferation and suppression of apoptosis. The cellular metabolism, surrounding microenvironment, and immune responses are altered as well [49,50,51]. Though, a crucial role in PDAC tumorigenesis, KRAS is highly refractory for direct inhibition. None of the direct KRAS inhibitors have given satisfying results so far. The need for different approaches is obvious. Targeting multiple KRAS effectors is currently under preclinical and clinical trials, and thoroughly reviewed elsewhere [52].

PDAC is characterized by high chemoresistance. One of the mechanisms includes the mutant KRAS activating the nuclear factor-erythroid 2–related factor 2 (NRF2), which is a central regulator of redox, metabolic, and protein homeostasis, among others, by altering glucose and glutamine metabolism and increasing glutaminolysis [53,54,55,56]. Mukhopadhyay et al. (2020) analyzed NRF2 expression in KRAS-driven pancreatic cancer tissues and cell lines. They found that a high NRF2 expression level was associated with a poor clinical outcome. Moreover, the authors showed that NRF2 regulates the sensitivity of PDAC cells to gemcitabine (a standard chemotherapeutic for PDAC), and concluded that targeting the NRF2-induced glutaminolysis by glutaminase inhibitors might sensitize PDAC cells to gemcitabine [57]. More studies are required to properly assess NRF2 expression in PDAC tissues and its influence on prognosis and gemcitabine resistance.

## 4. Molecular Subtypes

Detailed molecular characteristics of PDAC go beyond the scope of this review. Nevertheless, we will briefly outline the main trends in molecular subtyping.

In 2011 Collisson et al. described three subtypes (named classical, quasimesenchymal, and exocrine-like) and defined gene signatures for them. That study showed significant differences in patient outcomes and therapy responses between defined PDAC subtypes [58]. Moffitt et al. [59] (2015) studied PDAC gene expression in primary and metastatic tumors. They distinguished normal, tumor, and stroma-specific gene expression signatures highlighting the role of stroma in pancreatic cancer and they emphasized the need for analyzing it separately from the cancer cells. Authors suggested that the molecular characterization of the “quasimesenchymal” subtype defined in the study by Collisson’s team [58], was contaminated by the stroma of the tumor and the “exocrine-like” subtype by normal pancreatic tissue [59]. A total of four PDAC subtypes were defined with stroma-specific (normal and activated) and tumor-specific subgroups (basal-like and classical). A year after the report of Moffitt et al., Bailey et al. (2016) published results from a study that revealed four molecular subtypes: squamous, pancreatic progenitor, immunogenic, and aberrantly differentiated endocrine exocrine (ADEX), and described molecular pathways that are characteristic for each type [60].

Subtypes proposed in the studies briefly described above, seem to have overlapping features. Comparison of these was a subject of multiple studies that describe, in detail, the molecular nature of each group [15,61].

Recent studies have shown that PDAC tumors differ in the expression of immune features that are associated with the response to so-called immune checkpoint blockade (ICB) therapy. Liu et al. [62] analyzed immune signature gene sets including the activation of macrophage/monocytes, overall lymphocyte infiltration, TGF-β response, IFN-γ response (IFN-γ), and wound healing activity (wound healing) in the 383 pancreatic tumor samples. This allowed distinguishing between three subtypes (named C1–C3) of PDAC tumors, which differed in terms of the survival rates of patients. The authors highlight that a more personalized strategy should be considered when designing ICB treatment in PDAC patients [62].

## 5. Histopathologic Evaluation

Current guidelines for pathology reporting include recommendations from the International Collaboration on Cancer Reporting (ICCR) [19,63]. Since it is very important to follow these in daily pathologic workup, herein we particularly highlight some of them.

Multiple studies report that a standardized examination protocol for pancreatic tumors involving the head of the pancreas reveals a high positive margin (R1) rate (above 70%) [64,65,66,67]. It is important to evaluate all relevant surfaces including the anterior and posterior pancreatic surface, the surface of the superior mesenteric vein (SMV) groove, and the superior mesenteric arterial (SMA) dissection surface. A widely used standardized protocol for pancreatoduodenectomy specimens called LEEPP (Leeds Pathology Protocol) proposed by Verbeke et al. in 2006, relies on multicolor inking of all surfaces mentioned above and serial slicing of the whole pancreatic head specimen in an axial plane, perpendicular to the duodenum [68]. Additionally, trans-section margins including duodenal, stomach, bile duct, and pancreatic neck are recorded. Before the proposed standard, positive margins (R1) for pancreatoduodenectomy specimens were reported significantly less often, between 20% and 30% [69,70], and did not reflect proper prognostic value. Currently, a minimum clearance for R1 resection is considered 1 mm for trans-section margin and SMA/SMV dissection surfaces, whereas direct breaching of the surface is required for anterior/posterior pancreatic head involvement (0 mm) [19,71,72]. Patients with an R0 resection status have a significantly better prognosis but only when assessed with the LEEPP methodology [72].

The assumption of whether the incidence of low-grade or high-grade PanIN lesions in the trans-section margins can be considered a prognostic factor was disproved by Matthaei [73]. The risk of reoccurrence in R0 resected PDAC patients is not increased in such cases [22,73].

Reporting of the tumor histological subtype according to the WHO classification of tumors of the gastrointestinal tract, 5th edition, 2019 [16], is the core element in the ICCR guidelines for pathology reporting [19]. Other histological patterns and subtypes were shown to impact the patients’ outcomes, including the “large-duct pattern”, which could be easily misdiagnosed with IPMN-derived adenocarcinoma [74].

## 6. Morphological Heterogeneity

Pancreatic cancer (PC) is well known to be very heterogeneous in its molecular and morphological phenotype (Figure 2). It is one of the reasons for a poor patient prognosis, as current treatment options do not reflect the tumor heterogeneity and give insufficient results [18,75]. The mechanisms of failure are usually not well known, which compels researchers to look deeper into the molecular nature of these tumors. Numerous publications are explaining different aspects of genetic alterations in PC. There are, though, amazingly, few attempts to classify it in terms of morphological divergence yet this, from a pathological point of view, would give more practical information.

IPMNs are cystic tumors that can lead to invasive carcinomas. Generally, depending on the type of epithelial lining of the cystic structures, a gastric-type, intestinal-type, or pancreatobiliary-type are distinguished. Each has a different risk of progression into invasive PC [10,76].

Gastric-type IPMN is the most common (50–60%), with low-grade dysplastic epithelium and without MUC1 and MUC2 mucin expression on immunostaining (MUC1 negative, MUC2 negative). It progresses to invasive adenocarcinoma in 15% of cases. Intestinal-type IPMN accounts for about 20–30% of cases, 50% of them present with high-grade dysplasia, and immunostaining reveals MUC2 and CDX2 expression (MUC2 positive, CDX2 positive). The progression-to-PDAC rate reaches up to 40%. However, the vast majority progress to the invasive component in a form of colloid carcinoma, which has a slightly better prognosis. The third IPMN type, pancreatobiliary, is the rarest (10–15%), but most of them present as high-grade lesions with MUC1 expression (MUC1 positive). This type of IPMN is frequently invasive (60–70% of cases reveal invasive components) [10,16,76].

Another classification of IPMN tumors groups them by the place of origin, dividing them into “main-duct” and “branch-duct” lesions. Main-duct IPMNs present with high-grade dysplasia in 60% of cases and 45% of them are associated with invasive PC. Branch-duct IPMNs are mostly low-grade (only 25% have high-grade dysplasia), and only 20% reveal invasive components [77]. In 40% of cases, tumors are found to be multicentric [78].

**Figure 2 cancers-14-02321-f002:**
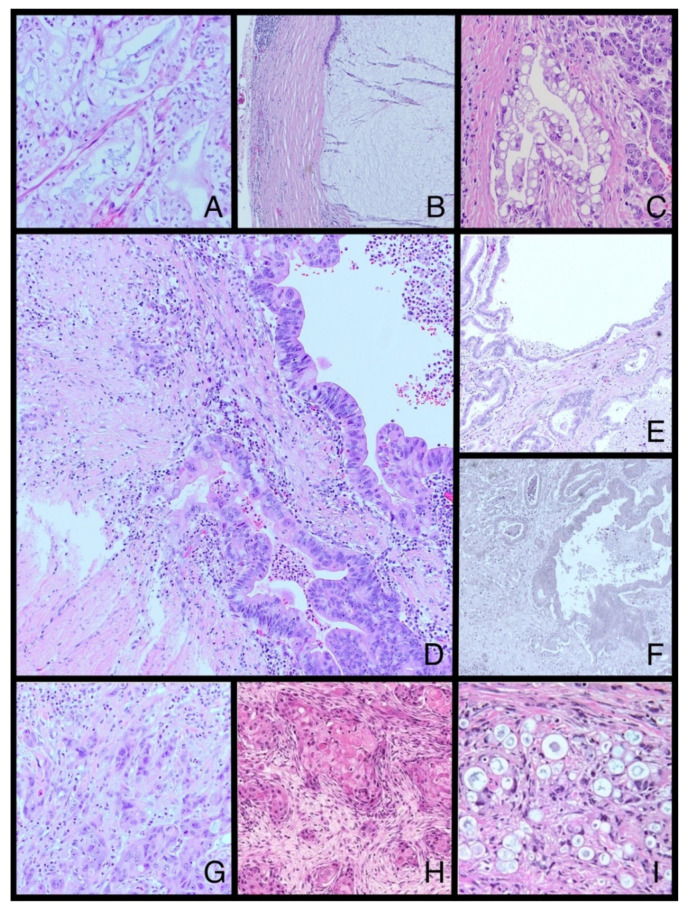
Selected WHO and non-WHO morphological subtypes of PDAC: (**A**)—clear cell subtype, (**B**)—colloid subtype, (**C**)—foamy gland pattern, (**D**)—PDAC derived from IPMN, invasive component, (**E**)—large duct/cystic papillary pattern, (**F**)—large duct/cystic papillary pattern negative for elastic stain, (**G**)—high-grade squamous differentiation, (**H**)—low-grade squamous differentiation—reproduced with permission from Kalimuthu et al., Gut, published by BMJ, 2020 [17], (**I**)—vacuolated cell pattern—reproduced with permission from Samad et al., Diagnostic Cytopathology, published by Wiley Periodicals, Inc., 2014 [79]; PDAC, pancreatic ductal adenocarcinoma; IPMN, intraductal papillary mucinous neoplasm; (**A**–**F**) orcein stain; original magnification (**A**,**C**,**D**,**G**–**I**) 100×, (**B**,**E**,**F**) 40×.

Muraki et al., in the single-institution study of 501 consecutive PDAC resected cases, pointed out that it is easy to misdiagnose IPMN-related carcinoma with IPMN-mimickers (pseudo-IPMN) [11]. As a pseudo-IPMN, the team described a secondary duct ectasia (retention cyst), large duct type PDAC, simple mucinous cyst, congenital cyst, paraduodenal wall cyst in grooves pancreatitis, and pseudocysts. Almost 3.8% of analyzed cases were classified as PDAC with pseudo-IPMN, compared to 6.2% with true IPMN (Table 1).

Pancreatic carcinoma may derive from an IPMN (PDAC derived from IPMN) or may develop apart from it (PDAC concomitant to IPMN). The histologic transition between IPMN and the invasive component should be revealed to determine the PDAC origin [63,80,81]. In the current WHO classification of pancreatic malignancies [16], there is no distinction between the two, and both should be reported as an “IPMN with associated carcinoma”. Although the different origin of IPMN concomitant carcinomas suggests a less favorable outcome, similar to that of conventional PDAC (cPDAC), in the work by Yamaguchi et al., both (derived from and concomitant) had similar, significantly favorable biological behavior [80]. The authors pointed out that this might be due to an earlier diagnosis of such cystic lesions.

Colloid carcinoma of the pancreas is characterized by the presence of large extracellular mucin pools (in at least 80% of the mass of the neoplasm) containing suspended neoplastic cells [16]. In the vast majority of cases, it is derived from intestinal-type IPMN. Multiple studies show that colloid IPMC had significantly better outcomes than tubular IPMC or cPDAC.

Another type described by the WHO classification [16] is a medullary carcinoma. A rare tumor of the pancreas that is often associated with microsatellite instability (MSI)/defective DNA mismatch repair (dMMR). Luchini and his team, in a meta-analysis of 34 studies, showed that the incidence rate of MSI/dMMR in PDAC is very low, ranging from 1–2%. It was significantly associated with medullary and colloid histological subtypes. Consequently, they suggested that cases of PDAC with medullary or colloid histology should routinely be examined in terms of MSI/dMMR, by use of immunohistochemistry [82].

Not many studies attempted to subtype a large group of “ductal adenocarcinoma, NOS”, concerning histological and immunohistochemical features. Kalimuthu et al. (2020) distinguished the morphological patterns of PDAC by separating two groups depending on gland formation (“gland forming” and “non-gland forming”) and correlated them with earlier described molecular subtypes (Figure 3) [17]. The four morphological patterns included conventional, tubulo-papillary, squamous, and composite. Nevertheless, this study did not explain PDAC morphological variability. A more comprehensive approach was demonstrated in a study by Sántha et al. [18]. From 233 foci selected from 39 pancreatic ductal adenocarcinoma specimens, the team analyzed 26 features including morphological and immunohistochemical patterns. Four common subtypes (67% of cases in the studied series) with significant differences in the areas of cancer cell proliferation (Ki67) and migration (collagen fiber alignment, metalloproteinases—MMP14), cancer stem cells (CD44, CD133, ALDH1), extracellular matrix (total collagen, collagen I and III, fibronectin, hyaluronan), cancer-associated fibroblasts (αSMA), and cancer–stroma interactions (integrins α2, α5, α1; caveolin-1) were distinguished. The patterns described as periglandular (PP), tendon-like (TP), fascicular (FP), and chicken wire (CP) were assessable by standard hematoxylin and eosin staining (H&E) and characterized by distinct features including the cancer cells, gland formation, and stromal compartment (Figure 3). There were significant differences between the subgroups including most of the features. What is important is that the study showed, among other things, heterogeneity in the stroma compartment composition that may affect different aspects of tumor growth, invasion potential, and resistance to therapy. The authors suggest that proper subtyping of pancreatic ductal adenocarcinoma may reveal these with better clinical outcomes and, secondly, allow the selection of subtypes that could benefit from new treatment options. It is important to give pathologists new ways of subgrouping “PDAC NOS” tumors that will be available for routine pathologic reporting and that will bring relevant and meaningful information to clinicians and patients.

Clear cell carcinoma of the pancreas is not a well-known entity and can be diagnostically misleading [83,84]. Kim et al., in the study of 84 pancreatic cancer specimens, reported that 24% of the analyzed cases contained significant clear cell components, and 14% of the studied cohort was defined as having clear cell carcinoma with over 75% of the tumor volume showing clear cell features [85]. In differentiating clear-cell-appearing tumors in the pancreas, other entities have to be taken into consideration, such as metastatic clear cell renal cell carcinomas, ovarian and adrenal carcinomas, and primary clear cell neuroendocrine tumors of the pancreas (frequently occurring in patients with von Hippel-Lindau disease) [86,87]. In the differential diagnosis, various histopathological stainings come in handy, including immunohistochemistry for carbonic anhydrase IX, HMB45, vimentin, PAX8, CD10, synaptophysin, or chromogranin [83]. Particularly useful for diagnosing clear cell pancreatic carcinoma of ductal origin (exocrine) is hepatocyte nuclear factor-1β (HNF1B), which significantly shows a stronger positivity more frequently in clear cell components compared to conventional ductal adenocarcinomas [85].

One more pattern called a “foamy gland pattern”, shows some similarities with clear cell carcinomas [88,89]. It has a benign-appearing look with well-formed glands and subtle infiltration. There is no data on whether these two entities are of a similar origin or follow the same molecular pathways. To the best of our knowledge, to date, no studies have compared clear cell and foamy gland patterns.

Another, non-WHO pattern of PDAC called “large duct pattern” (also called “cystic papillary pattern”), can be misdiagnosed as an IPMN [74]. In this pattern, carcinomatous ducts are enlarged over 0.5 mm. If at least 50% of the tumor glands show such dilatation, the name “large duct carcinoma” is used. In some cases, an elastic fiber staining (i.e., orcein stain) helps to distinguish between non-malignant (ducts dilated due to occlusion) and malignant ducts. Large duct carcinoma is frequently accompanied by perineural invasion (88%) [90]. There are no mucin pools and signet ring cells, in contrast to colloid carcinoma. A total of 73% of large duct PDACs stain with MUC1 positively and over 55% for MUC6 [74,91,92], which might be helpful to differentiate it from gastric-type IPMN. The term “cystic papillary pattern” is sometimes distinguished from a large duct pattern, describing more complex or papillary structures [92]. However, some authors suggest that both represent the same entity, reflecting distinct evolutionary stages [90]. Interestingly the large duct pattern of PDAC might fit into the subgroup distinguished by Kalimuthu et al. called “tubulo-papillary”, rather than the “gland forming” group [17].

## 7. Immunostaining

Currently, there is no immunostaining marker that would be recommended for the routine pathological diagnostic workup of PDAC. Nevertheless, there have been multiple reports suggesting that p53 [93,94,95] and insulin-like growth factor-II mRNA-binding protein 3 (IMP3/IGF2BP3/KOC) [96,97,98,99,100,101,102,103] could significantly help to avoid misdiagnosis. Overexpression of these markers correlates with patient prognosis. Liu et al. reported 90% of PDAC cases to be positive for IMP3, Maspin, and S100 calcium-binding protein P (S100P) expression [104]. Another study utilized IMP3, Maspin, S100P, and von-Hippel-Lindau gene protein (pVHL) for comparison between autoimmune pancreatitis (AIP), PDAC, and normal pancreas specimens [105]. The authors showed that although weak and focal expression was seen in AIP and a normal pancreas, PDAC was characterized by strong overexpression of Maspin, IMP3, and S100P in 95%, 75%, 75%, respectively, whereas weak and no expression was seen in 0%, 25%, 10% of PDAC cases, respectively. Furthermore, 100% of studied PDAC samples were negative for pVHL. Recently, Senoo et al. [106] and Mikata et al. [107] conducted retrospective and prospective studies evaluating the usefulness of p53 and IMP3 in endoscopic ultrasound-guided fine needle aspiration (EUS-FNA) specimens of pancreatic tumors. They reported that none (0%) of the benign lesions analyzed expressed IMP3 nor p53, whereas, for malignant lesions, IMP3 was positively expressed in 60% (69% in PDAC) cases and 49% of them were characterized by overexpression of p53.

Of note, as mentioned above, hepatocyte nuclear factor-1β (HNF1B) shows a strong positive expression in clear cell pattern ductal adenocarcinoma of the pancreas [85].

CDX2 is known to be a driver for gastric-to-intestinal type progression in IPMN [47]. In 95% of analyzed intestinal-type IPMN cases, CDX2 showed strong positive nuclear staining in more than 90% of cells. Xiao et al. reported a loss of CDX2 expression during PanIN progression from low-grade to high-grade lesions, and only one-third of PDAC samples showed weak CDX2 expression [108]. It is believed that colloid carcinoma of the pancreas develops almost exclusively through the intestinal-type IPMN progression pathway [10]. Further research would be required to assess the prevalence of CDX2 expression strictly in pancreatic colloid carcinoma.

P63 and p40 stain squamous components of adenosquamous carcinomas. This entity presents with different molecular bases (“basal-like” [59] or “squamous” [60] molecular subtypes) and a slightly worse prognosis than cPDAC [17,18,59,60,109]. Recently, Haugk et al. showed that p40 immunostaining was a useful tool in detecting squamous differentiation of PDAC in fine needle biopsy (FNB) samples [110].

Leukemia inhibitory factor (LIF) and interleukin-6 (IL-6) are potentially promising biomarkers for early diagnosis of PDAC, that can be detected in patients’ serum [12,13]. Aside from the LIF serum levels, a raised LIF concentration was observed in pancreatic cancer tissue samples compared to chronic pancreatitis or benign lesions (PanIN) and inversely correlated with the tumor differentiation level [13,111,112,113].

The immunohistochemical staining markers of malignancy and those with prognostic relevance are summarized in Table 2.

## 8. Cancer–Stroma Interactions

PDAC invasion is characterized by an extensive, dense, desmoplastic stroma that is not only a silent actor but plays a crucial role in the tumor growth, maintenance, invasion, metastatic potential, and chemoresistance [130,131,132,133,134,135,136,137,138,139]. The complicated relationship between cancer cells and the stroma was metaphorically, yet very vividly envisaged by Adamek and Stoj (2014). They have proposed a figurative concept of cancer as a form of the “mafia” within the body, in which the cancer cells “corrupt” non-neoplastic cells and, as a result, aid and abet them in “the crime of cancer”. What is more, the “criminal cells” may even cunningly change their properties and mislead researchers, altering the study results and efficacy of treatment [140]. This holistic approach to cancer biology explains some difficulties in the cancer–stroma interplay research.

The explanation of the complex interactions between pancreatic cancer cells and the stroma compartment will undoubtedly be a milestone in the development of PDAC therapy strategies. Currently, multiple trials are investigating possible options in targeting stromal compartment mechanisms or cancer-to-stroma interaction pathways with varied preliminary reports [9,136,141,142,143,144,145,146]. There are strong clinical implications of the presence of different stromal compartment composition features. Some authors suggest that reporting them should be a part of the routine pathological workup [18].

The research on PDAC stromal nature, particularly the cancer-associated fibroblasts (CAFs) that are the main component of the tumor microenvironment (TME), faces many difficulties, caused by, among other things, inconsistencies in key definitions. In 2019, a Banbury Center meeting of international researchers and clinical scientists gathered in New York (USA) and agreed on a consensus statement where they summarized good practice advice and described the recommended methodology for CAF research [147].

In the normal pancreas, fibroblasts, called pancreatic stellate cells (quiescent—qPSCs), are producers of connective tissue and play a substantial role in tissue repair processes. In case of damage (chronic pancreatitis) or carcinogenesis, qPSCs become activated and starts presenting a myofibroblast-like phenotype that expresses α-smooth muscle actin (α-SMA) [148]. Activated PSCs (aPSC) secrete over 13 times more proteins responsible for the proliferation, inflammation, extracellular matrix remodeling, cell motility, and invasion (641 by aPSCs vs. 46 by qPSCs) [149]. It is believed that aPSCs are the main source of CAFs [147,150,151]. Other important sources include bone marrow-derived mesenchymal stem cells (MSCs) [152,153,154,155,156,157]. Some studies raise the possibility of adipocyte and pericyte conversion into CAFs, but the results are ambiguous [158,159,160,161]. Multiple factors play a role in PSC activation. As mentioned earlier (see Section 3), PanIN progression mechanisms involve mutations in tumor-suppressing genes that lead to the activation of TGFβ signaling pathways (SMAD4/TGFBR1/TGFBR2). Loss of p53 activates Janus kinase/signal transducer and activator of transcription (JAK/STAT) signaling pathway, which leads to tumor initiation and progression, by the stimulation of CAFs/PSCs, among others [162,163,164]. Additionally, CAFs secrete paracrine factors, that act as STAT3 activators on cancer cells, of which, leukemia inhibitory factor (LIF) is a key player. Other CAF-producing growth factors include TGFβ, growth arrest-specific protein 6 (GAS6), fibroblast growth factor 5 (FGF5), growth differentiation factor 15 (GDF15), and hepatocyte growth factor (HGF) [111,165,166]. Other secretory mechanisms of CAFs promote angiogenesis [167,168] or act as an immunosuppressor [169,170], but results for the latter seem to be inconsistent [147,171].

The catabolic mechanisms of CAFs modulate cancer cell metabolism and fuel cancer, with energy sources like amino acids and other nutrients, sustaining the tumor growth [172,173,174,175] in a process with the suggested name “reversed Warburg effect” [176,177,178]. From one side, CAFs produce a collagen-rich ECM, from the other, cancer cells stimulate CAF autophagy that “produces” a substantial dose of alanine [179].

Multiple studies recognized CAFs within the pancreatic cancer stroma to be a heterogeneous population of cells that present divergent phenotypes and seem to have different roles. Three main groups are myofibroblastic (myCAFs), inflammatory (iCAFs) and antigen-presenting CAFs (apCAFs) [150,180,181,182,183]. MyCAFs are located in the periglandular region of the cancer site and express high levels of α-SMA and low levels of IL-6. Activation (PSCs to myCAFs) is believed to be rendered via the TGFβ/SMAD signaling pathway. More distally to the glands, iCAFs are found. They express high levels of IL-6 (low α-SMA), due to IL-1/LIF/JAK/STAT pathway activation [135]. The third subtype, called antigen-presenting CAFs, shows high expression of MHC II family genes but their activation mechanisms and specific features are yet to be determined [180].

Classic biomarkers used for CAF detection include α-SMA, fibroblast activation protein (FAP), fibroblast-specific protein 1 (FSP1/S100A4), platelet-derived growth factor receptors (PDGFRα, PDGFRβ), or podoplanin (PDPN/gp38). However, some studies recognize the heterogeneity of CAF expression profiles [184,185]. The Banbury Center meeting [147] highlighted the importance of determining specific subtypes of CAFs in the tumor stroma with the use of new immunohistochemistry methods but, currently, there is not enough evidence to state a consensus in this field.

Stroma composition was a part of the molecular subtyping by Moffitt et al. [59]. Authors distinguished stroma-specific from the tumor-specific gene expression and revealed two molecular subtypes regarding the stroma compartment, namely “activated” and “normal”. Stroma evaluation played a substantial part in the morphological subtyping of PDAC specimens by Sántha et al. [18]. They studied such components as an extracellular matrix with collagen composition and arrangement, fibronectin and hyaluronan deposition, cancer-associated fibroblasts (α-SMA staining and collagen density with the use of the so-called activated stroma ratio [114,115]), and cancer–stroma interactions (integrins α2, α5, β1, metalloproteinases—MMP14 and caveolin-1). The biological role of all of these features was documented substantially in publications prior to that study. Proposed morphological subtypes that presented with significantly different phenotypes regarding the above, showed the importance of stroma compartment evaluation in pathological practice.

Cancer stem cells (CSCs) represent a small number of cancer cells within the tumor that define its potential to grow and propagate. CSCs have stem cell properties and tend to be self-renewable, and multipotent. Some studies showed that CSCs might be responsible for tumor initiation, rapid growth, resistance to therapy, recurrence, and metastases [186,187,188,189]. The induction of CSCs is related to the epithelial-to-mesenchymal transition (EMT) regions [190,191,192,193]. In pancreatic cancer, CSCs express CD24, CD44, CD133, aldehyde dehydrogenase 1 (ALDH1), and epithelial-specific antigen (ESA) [186,188,193]. Pancreatic stromal stellate cells (PSCs) cooperate in a paracrine manner with the CSCs to increase their invasiveness and self-renewal properties via the Nodal/Activin signaling pathways [194,195]. For some authors though, the idea of CSCs is controversial [196].

Detection of CSCs in pathology reporting might be of importance due to potential therapy options [195] and correlation with clinical prognosis [189,193]. Sántha et al. [18] studied the expression of CD44, CD133, and ALDH1 in pancreatic cancer cells as part of a morphological subtyping scheme and found significant differences in marker expression between proposed subtypes that correlated with other morphological and immunohistochemical features.

## 9. Prognosis

Summarized information about histomorphologic features of PDAC is presented in Table 3. The overall 5-year survival rate for pancreatic ductal adenocarcinoma is less the 10% [1]. The majority of patients present with inoperable and non-curable tumors. The survival rate, among other aspects, depends on the stage of the tumor at the time of diagnosis. A total of 10% of patients had T1-T2 disease with a 5-year survival rate reaching 32%, while the rate dropped to 12% for T3 tumors. More than half of the patients had T4 stage tumors with a 5-year survival rate of 3% [1].

The prognosis for patients with cancers derived from intraductal papillary mucinous neoplasm (IPMC), compared to conventional PDAC (cPDAC), has been in debate for a long time, sometimes with conflicting results. It is important when analyzing those results to bear in mind that often the distinction between “derived from” vs. “concomitant to” was not properly addressed [197,198,199,200,201]. Poultsides et al. reported significantly better survival only for tumors that did not reveal known adverse prognostic factors, such as poor differentiation, involved surgical margins (R1), or vascular (LV1), or perineural invasion (PNI) [202]. Okabayashi et al. [203] published results showing that invasive carcinomas derived from branch-duct IPMNs might be more aggressive with poorer patient outcomes compared to those derived from main-duct IPMNs. Notwithstanding, the risk of malignant transformation in main-duct IPMN is higher than in branch-duct IPMN. Additionally, SMAD4 and TGFβ expression was significantly increased in the carcinomas derived from branch-duct IPMNs. The question regarding a better prognosis of IPMC vs. cPDAC, regardless of the stage of the tumor, remains unanswered. What is certain is that IPMC patients are diagnosed earlier, which gives them a better start.

Multiple studies show that colloid carcinoma has a beneficial outcome compared with conventional subtypes (including tubular IPMC). The 5-year survival rates ranged from 57–87% to 24–55% [199,200,202,204,205,206]. Additionally, a lower incidence rate of adverse factors (PNI, LV1, R1, nodal involvement, poor differentiation) has been shown.

The patient prognosis in large duct carcinoma seems to be similar to that in tubular carcinoma or slightly better, probably because of good differentiation occurs more frequently [74,90,92]. Nevertheless, it is important to distinguish it from other cystic lesions, like IPMNs.

Surprisingly, unlike in other locations (colorectal, gastric, duodenal, or ampullary cancers) [207], medullary carcinoma of the pancreas is not correlated with a better prognosis [82], but there is a very small amount of data available. The KEYNOTE-158 clinical trial, including patients with MSI/dMMR PDAC tumors, showed objective responses in only four out of 22 patients (one complete, three partial) included in the trial [208].

Pancreatic adenosquamous carcinoma (ASPC) tends to have a worse outcome compared to ductal adenocarcinoma, NOS [129]. The mechanisms of squamous differentiation of pancreatic cancer cells are not yet fully recognized [209], but recent studies showed p6transcription factor to be a master regulator of this pathway [209]. Lately, Kaiser et al. [109] conducted a study with 91 ASPC cases from a total of 4009 patients who underwent surgery. The results showed that median post-surgical survival in ASPC compared to PDAC was 10.8 vs. 20.5 months. Another study reported a lower 2-year survival rate in ASPC vs. PDAC (29% vs. 36%) [210].

**Table 3 cancers-14-02321-t003:** Summary of histomorphologic features with prognostic importance. 5-YSR, 5-year survival rate; 2-YSR, 2-year survival rate; cPDAC, conventional pancreatic ductal adenocarcinoma; IPMC, IPMN derived carcinoma; PNI, perineural invasion; LI, lymphovascular invasion; VI, venous invasion; R1, positive surgical margins; NI, nodal involvement; PANR, PDAC-associated neural remodeling; LEEPP, Leeds Pathology Protocol; SMV/SMA, superior mesenteric vein/superior mesenteric artery; HG PanIN, high-grade pancreatic intraepithelial neoplasia.

**Tumor size (incidence)**	
T1-2 (10%) T3 T4 (>50%)	smaller tumor size—better prognosis (5-YSR) [1]: T1-2 => 32% T3 => 12% T4 => 3%
**Histologic type**	
IPMC	results are ambiguous [197,198,199,200,201,202] patients are diagnosed earlier [80]
Colloid	better prognosis than cPDAC (5-YSR) [199,200,202,204,205,206]: 57–87% to 24–55%; significantly lower incidence of PNI, LI, VI, R1, NI, or poor differentiation
Large duct/Cystic papillary	same prognosis as cPDAC or slightly better because of a good differentiation [74,90,92]
Medullary	not a better prognosis, in contrast to other sites of digestive system [82,207]
Adenosquamous (ASPC)	ASPC vs. cPDAC (2-YSR) [210]: 29% vs. 36%
**Other features**	
Lymphatic and venous invasion	both should be reported separately because they represent different biological processes—lymph node metastasis and distant, blood-borne spread [19] considered as a non-core element due to possible difficulties in distinction; elastin staining might be helpful [19]
Lymph node status	the mechanism of lymph node involvement should be recorded, as direct or metastatic [211,212,213,214] prognostic influence of direct NI is similar to that of nodal negative disease
Perineural involvement	adverse prognostic factor [19,215,216,217,218] prognostic influence of the extent of PNI and PANR is yet to be determined
Surgical margin status	positive margin (R1) found in >70% of cases [19] correlates with survival, but only when assessed with use of standardized examination protocol that includes evaluation of all surfaces (LEEPP) [72] trans-section margins and SMV/SMA dissection surfaces positive when <1 mm anterior and posterior dissection margins positive when directly involved [19,71,72]
High-grade PanIN	HG PanIN in transection margin is not considered a prognostic factor [22,73]

Kalimuthu et al. suggested that morphological classification is a better prognostic factor than the standard three-tiered grading system, as most of the studied cases were classified as moderately differentiated. They grouped specimens into two groups, defined as group A—which showed less than 40% “non-gland forming” components, and group B—with more than 40%. Group A had significantly better overall survival than group B [17].

## 10. Immunohistochemical Prognostic Factors

Loss of p16 (CDKN2A) expression in cancer cells was significantly associated with lymphovascular invasion and metastatic disease [95,128].

The prognostic value of SMAD4 aberrations is ambiguous. Some reports associate the loss of SMAD4 with poor survival and early metastasis [95,122,123], but some did not achieve similar results with significant relevance [124,125,126,127].

Song et al., in a study of 62 PDAC cases, showed that Janus kinase 2 (JAK2) immunostaining is an independent poor survival factor [121] though, there is a limited amount of data regarding this pathway (JAK/STAT) biomarker expression. There are some new potential treatment options and patients might benefit from JAK/STAT biomarker reporting in the future [219,220,221,222].

Some studies have shown that podoplanin-positive CAFs in the PDAC stroma compartment were associated with poor prognosis, aggressive behavior, and larger tumor size [118,119,120].

One report suggested, that Meflin-positive CAFs in PDAC stroma prevents poor differentiation of the tumor and are markers of a favorable outcome [117]. Another study by Ikenaga and colleagues showed that CD10-positive CAFs were associated with positive lymph node metastasis and a shorter survival time [116]. The ratio of the α-SMA-stained area to the collagen-stained area was defined as the activated stroma index (ASI). Erkan et al. [114] differentiated fibrolytic (high α-SMA/low collagen), fibrogenic (low α-SMA/high collagen), inert (high α-SMA/high collagen), and dormant (low α-SMA/low collagen) pancreatic stroma composition. Significant differences in patient outcomes and progression-free survival between these composition types have been reported [114,115].

Last, but not least, a strong expression of hepatocyte nuclear factor-1B (HNF1B) in PDAC cases was correlated with a worse prognosis regardless of the morphological features [85]. See Table 2 for summarized info about immunohistological features of PDAC with prognostic relevance. 

## 11. Perineural Invasion

Pancreatic cancer is characterized by early and extensive perineural invasion (PNI). Studies have shown that PNI in PDAC is an independent poor prognostic and an early recurrence factor [215,216,217,218]. It is considered a core element in pathology reporting [19]. Recent studies highlight the active role that nerves play to facilitate tumor spread [223,224,225,226,227,228,229]. Nervous cells interplay with cancer cells and the stromal compartment cells (CAFs, PSCs, and tumor-associated macrophages—TAMs) [230]. PDAC-associated neural remodeling (PANR) is a proposed term describing the alterations in the nerve compartment caused and facilitated by PDAC tumors, and resulting in higher nerve densities in PDAC due to peripheral nerve fiber infiltration and axonogenesis [112]. Bressy with colleagues [112] showed that PANR was supported by leukemia inhibitory factor (LIF). LIF influences Schwann cells and dorsal root ganglia (DRG) neurons via modulation of the JAK/STAT signaling pathways and facilitates their migration and differentiation. They suggested that the use of LIF-inhibitors might suppress PANR, limit tumor spread, and increase patients’ quality of life [112].

More studies are required to comprehensively assess the PNI prognostic influence and to explain the divergence of PNI levels among PDAC tumors.

## 12. Lymph Node Metastasis

Regional lymph node status is a well-known poor prognostic factor in PDAC and it is considered a core element in the ICCR guidelines for pathology reporting [19]. However, the 8th edition of the TNM classification of the American Joint Committee on Cancer (AJCC) does not distinguish the mechanism of lymph node invasion (LNI). In 2015 Williams et al. [214] examined PDAC specimens regarding the mechanism of LNI (distinguishing true “metastatic” spread and “direct” LNI—Figure 4) and compared patients’ survival rates. They concluded that “direct”-only LNI patients had a similar overall survival to those with node-negative disease. Recently, other authors reported similar results [211,212,213].

## 13. Early Diagnostic Options

There are no efficient early diagnosis tools for pancreatic carcinoma. Late-stage disease at diagnosis is certainly a major issue that partially leads to overall poor survival rates.

Serum carbohydrate antigen (Ca19-9) is used in the diagnostic work-up of patients being diagnosed with pancreatic tumors, but it is neither specific nor sensitive for malignant lesions [231,232]. Some studies showed better usefulness of assessing IL-6 serum levels in differentiating PDAC patients from chronic or acute pancreatitis [233,234,235]. Recently, LIF was reported to be a promising serum biomarker of pancreatic malignancy [112], as much as a metastatic disease predictor for PDAC patients [236] and a therapy response monitor [13,111]. Moreover, LIF was shown to be a good biomarker for immune checkpoint blocker (ICB) therapy efficacy, which is a novel immunotherapeutic option for patients with solid tumors [237]. Loriot et al. identified elevated LIF serum levels as a poor prognostic factor for ICB-receiving patients [238].

The development of new diagnostic technologies might shed new light on the PDAC early diagnostic field. Raman spectroscopy (RS) and surface-enhanced Raman spectroscopy (SERS) [239,240] methods were used for detecting earlier untraceable amounts of biomarkers in PDAC patients’ serum [241,242]. SERS with a plasmonic gold nanohole array was used for the detection of DNA methylation aberrations [243]. These innovative methods might increase the sensitivity of aberrant methylation marker detection in the circulating cell-free DNA (cfDNA) and circulating tumor cells (CTC) in PDAC patients [244,245] (Table 4).

## 14. Molecular Characteristics of Malignant Pancreatic Tissues

A growing number of scientific articles confirm an important role of molecular spectroscopy in the characteristics of the chemical structure and composition of various malignant tissues [246,247,248,249] Due to high chemical selectivity, both Raman and infrared spectroscopies can become efficient tools supporting the molecular screening of pancreatic tissue sections. This methodology provides information about the content of various biologically significant molecules and functional groups, including phospholipids and triglycerides, proteins, nucleic acids, phosphates, and carbohydrates. The results indicate differences in the metabolic pathways typical for various neoplasms. The main advantage of the molecular spectroscopic approach is achieving information about samples in a label-free and noninvasive manner. The research potential of spectroscopic methods has not yet been fully explored in the investigation of pancreatic cancer (Figure 5) [250,251].

Tissues are complex systems and to achieve a complete overview of their molecular structure, hyperspectral mapping, which provides full spectral information from each pixel, is applied. To reduce data dimensionality and extract the most important information—marker bands of molecular pathologies, from the acquired data, various methods of multivariate data analysis are used [252]. K-means clustering (KMC), and principal component analysis (PCA) are commonly performed to explore spectral variation in maps acquired from tissue sections [253,254,255].

Molecular spectroscopy coupled with multivariate data analysis support standard immunohistochemical and histological staining-based procedures. The comprehensive approach may increase the effectiveness of proper diagnoses of pancreatic and ampullary cancer and their subtyping [251].

## 15. Conclusions

The field of pancreatic cancer research is rapidly evolving. Reports of newly revealed pathomechanisms of the nature of these tumors are published daily. Nevertheless, many aspects of pathologic evaluation are still uncertain (Table 5). Although it is sometimes hard to stay on top of things with such a dynamically increasing amount of knowledge, it is crucial to be able to pull out practical information that impacts the diagnostic process, called a pathologic evaluation. Hopefully soon, we will witness a great change in PDAC patients’ prognosis, whether through the development of new early screening methods or new therapeutic options. In the meantime, pathologists should do whatever they can to make the pathologic reporting meaningful for clinical and research purposes.

## Figures and Tables

**Figure 1 cancers-14-02321-f001:**
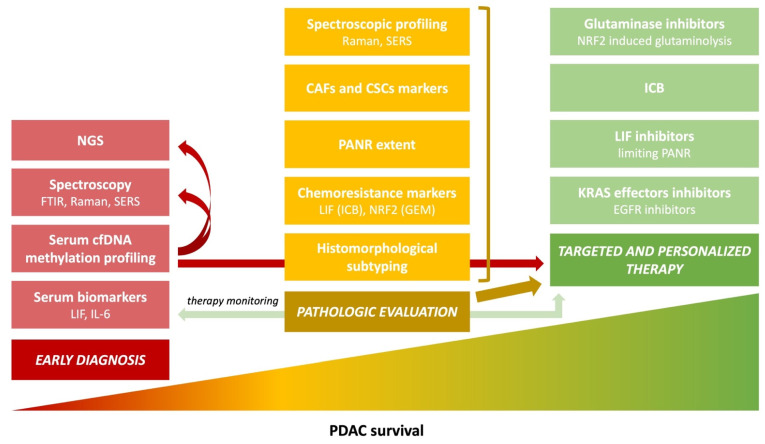
Main trends in PDAC pathology and research that are expected to improve survival. Poor PDAC patients’ prognosis is multifactorial—no sensitive and specific early diagnostic methods is one of the reasons. Another is the resistance to available therapeutic options, which is caused, among other things, by the tumor’s molecular and morphological heterogeneity. Detailed pathological reporting is crucial for targeted and personalized therapy. The development of new diagnostic methods, combined with a proper pathologic evaluation and spectroscopic profiling, leads to effective treatment. Altogether, this will increase PDAC patients’ survival rates. PDAC, pancreatic ductal adenocarcinoma; LIF, leukemia inhibitory factor; IL-6, interleukin-6; cfDNA, cell-free DNA; FTIR, Fourier transform infrared spectroscopy; Raman, Raman spectroscopy; SERS, surface-enhanced Raman spectroscopy; NGS, next-generation sequencing; ICB, immune checkpoint blockers; NRF2, nuclear factor-erythroid 2–related factor 2; GEM, gemcitabine; PANR, PDAC-associated neural remodeling; CAFs, cancer-associated fibroblasts; CSCs, cancer stem cells; EGFR, epithelial growth factor receptor.

**Figure 3 cancers-14-02321-f003:**
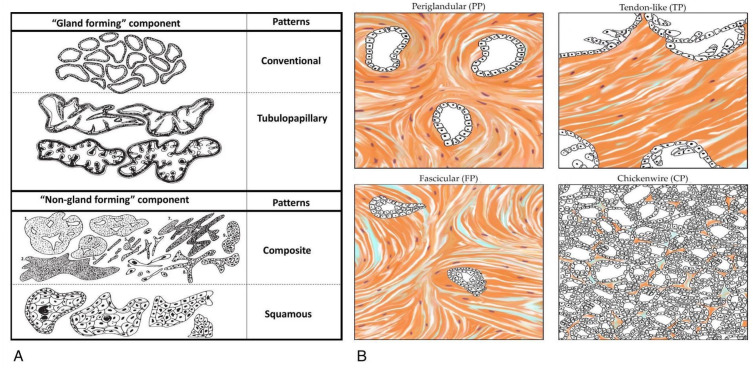
Morphological classifications: (**A**)—proposed by Kalimuthu et al. [17], and (**B**)—proposed by Sántha et al. [18]. See the text (Section 6) for further details on the topic. Images are reproduced with permission from Kalimuthu et al., Gut, published by BMJ, 2020 [17] and from Sántha et al., Cancers, published by MDPI, 2021 [18].

**Figure 4 cancers-14-02321-f004:**
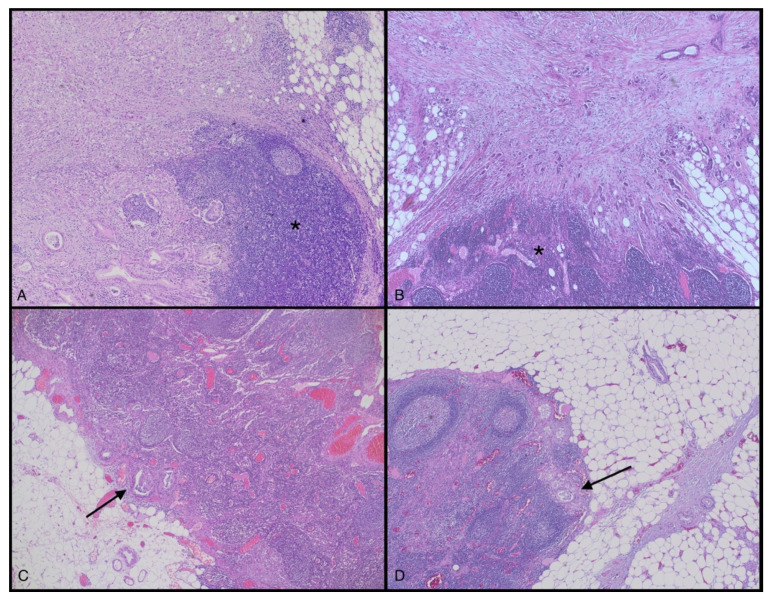
Direct lymph node involvement by PDAC (NI): Almost the whole left part (**A**) and more than an upper two-thirds (**B**) are occupied by solid PDAC growth, which directly invades the remnants of a lymph node (asterisks). This form of NI in PDAC has a different prognostic significance compared to a true metastatic spread (arrows in (**C**,**D**)). See the text (Section 12) for further details on the topic. (H&E stain, original magnification ×4).

**Figure 5 cancers-14-02321-f005:**
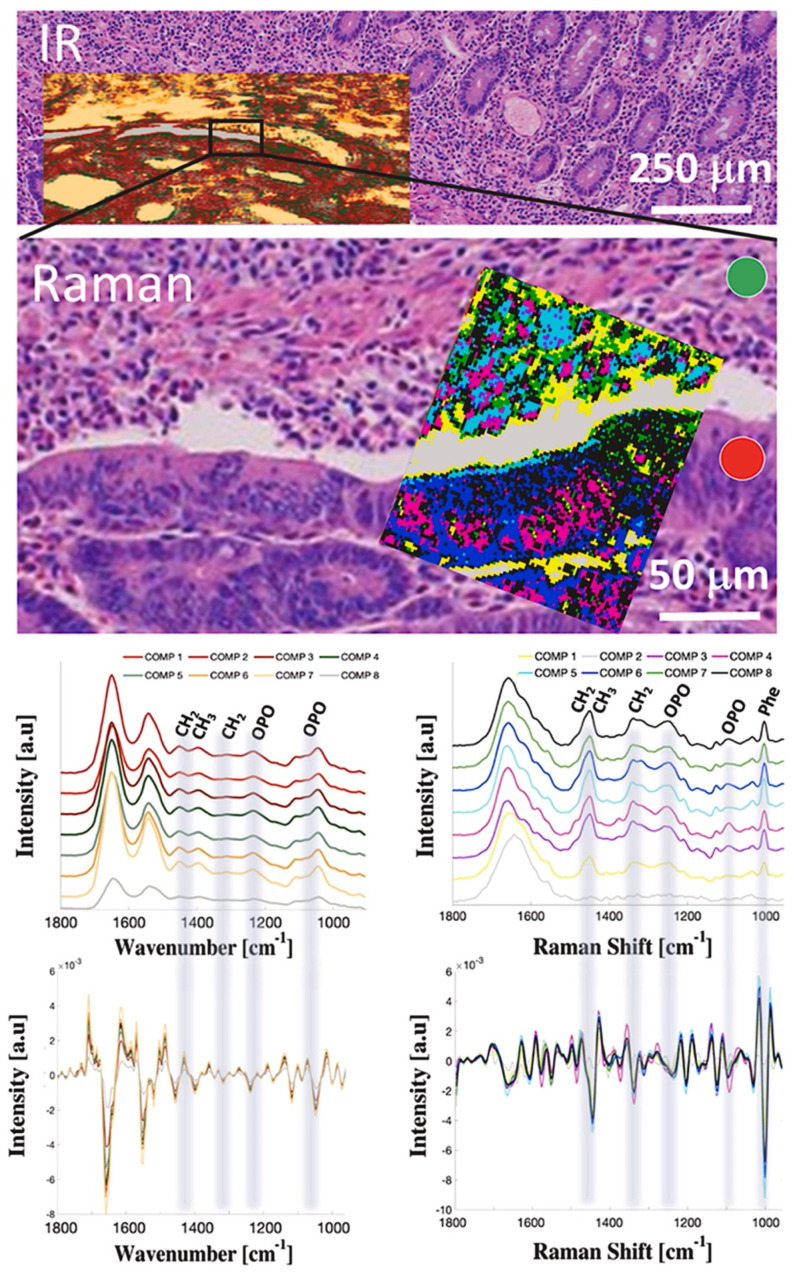
Spectroscopic mapping of ampullary adenocarcinoma: A hematoxylin-eosin slide image of ampullary cancer tissue with superimposed FTIR and Raman hyperspectral maps treated with hierarchical cluster analysis, and averaged spectra with corresponding second derivatives from each cluster; spectroscopic maps cover both cancerous (red circle) and noncancerous—stroma (green circle) tissue fragments. Reproduced with permission from Szymonski et al., Clinical Spectroscopy, published by Elsevier B.V, 2021 [251]. FTIR, Fourier transform infrared spectroscopy.

**Table 1 cancers-14-02321-t001:** The differential diagnosis for IPMN-associated carcinoma. PanIN, pancreatic intraepithelial neoplasia; PDAC, pancreatic ductal adenocarcinoma; IPMN, intraductal papillary mucinous neoplasm.

Pseudo-IPMN	Differential Pathology Findings
General	usually small cystic component (typically <2 cm) and large invasive component
Retention cysts (secondary duct ectasia)	located in upstream areas of the tumor round, unilocular, usually with low-cuboidal lining and open lumina; no contour irregularities; no papillary configuration; no classical cell types possible PanIN lesions or colonization with carcinoma cells by an intraductal spread from invasive carcinoma
Simple mucinous cysts	unilocular and round; smooth internal lining composed of simple mucinous lining with occasional folding and atypia; no florid papilla formation low-grade PanIN common
Congenital cysts	relatively large cysts (6–11 cm) outer muscular coat; lining by a relatively normal ciliated/respiratory, gastric, or intestinal mucosa invasive carcinoma may arise directly from the lesion
Paraduodenal wall cyst	partially lined by granulation tissue (pseudocyst) and partially by columnar mucinous epithelium usually, a decade older male patients, suggesting that invasive carcinoma develops as a long-term consequence of a chronic pancreatitis
Large-duct and cystic papillary pattern of PDAC	grossly, multiple cystic structures, some >1 cm within the PDAC irregularly distributed large ducts with jagged edges; lined by columnar mucinous cells often having deceptively bland cytological features; variable degrees of papillomatosis negative for elastic fibers stains (orcein stain)
PDAC concomitant to IPMN	carcinomas derived from and concomitant to IPMN should be distinguished the histologic transition between IPMN and the invasive component should be revealed to determine the PDAC origin [76,77,78]

**Table 2 cancers-14-02321-t002:** Summary of immunohistochemical staining features of PDAC. pVHL, von-Hippel-Lindau gene protein; IMP3, insulin-like growth factor-II mRNA-binding protein 3; S100P, S100 calcium-binding protein P; LIF, leukemia inhibitory factor; ALDH1, aldehyde dehydrogenase 1; CP, chronic pancreatitis; HNF1B, hepatocyte nuclear factor-1B; JAK2, Janus kinase 2; CAFs, cancer-associated fibroblasts; α-SMA, alpha-smooth muscle actin; NRF2, nuclear factor-erythroid 2–related factor 2.

**Markers of malignancy**	
pVHL [105]	loss of pVHL was found in 100% of PDAC samples
Maspin [105]	strong overexpression in 95% of PDAC samples weak and none in 0% of PDAC samples
IMP3 [105]	strong overexpression in 75% of PDAC samples weak and none in 25% of PDAC samples
S100P [105]	strong overexpression in 75% of PDAC samples weak and none in 10% of PDAC samples
p53 [106,107]	expressed in 0% of benign lesions overexpressed in 49% of malignant lesions
LIF [13,111,112,113]	elevated LIF concentration was observed in PDAC stroma compared to CP or benign lesions expression inversely correlated with the tumor differentiation level
**Markers with prognostic relevance**	
CD44, CD133, ALDH1 [18]	markers of cancer stem cells correlates with poor survival, early recurrence, and metastasis gemcitabine treatment monitoring
HNF1B [85]	strong expression is correlated with worse prognosis independent of morphology
Activated Stroma Index (ASI) [114,115]	α-SMA-stained area to the collagen-stained area of tumor stroma ASI was an independent survival predictor, comparable with nodal status lowest ASI (fibrogenic => low α-SMA, high collagen)—best prognosis highest ASI (fibrolytic => high α-SMA, low collagen)—worst prognosis
CD10 [116]	stains CD10-positive CAFs in the tumor stroma positive lymph node metastasis and shorter survival time
Meflin [117]	stains Meflin-positive CAFs in the tumor stroma favorable outcome
Podoplanin [118,119,120]	stains podoplanin-positive CAFs in the tumor stroma poor prognosis, aggressive behavior, and larger tumor size
JAK2 [121]	independent poor prognostic factor
SMAD4 [95,122,123,124,125,126,127]	results ambiguous
p16 [95,128]	loss of p16 expression was correlated with lymphovascular invasion and metastatic disease
p63, p40 [129]	stains squamous differentiation poor prognostic factor
NRF2 [57]	strong expression was found to be a worse prognosis factor and corelate with gemcitabine resistance more studies are required for proper prognostic influence assessment

**Table 4 cancers-14-02321-t004:** New early serum diagnostic methods. Ca19-9, carbohydrate antigen; IL-6, interleukin 6; LIF, leukemia inhibitory factor; cfDNA, cell-free DNA; CTC, circulating tumor cells; SERS, surface-enhanced Raman spectroscopy; NGS, next-generation sequencing.

Pancreatic Malignancy Serum Biomarkers	
**Ca19-9**	not specific, nor sensitive for malignant lesions [231,232]
**IL-6**	better usefulness than Ca19-9 [233,234,235]
**LIF**	promising serum biomarker of pancreatic malignancy [112] metastatic disease predictor [236] therapy response monitoring [13,111] good biomarker for immune checkpoint blocker (ICB) therapy efficacy [237,238]
**cfDNA methylation**	novel sensitive techniques, but yet under development and research [245] NGS and SERS used for cfDNA methylation markers [244]

**Table 5 cancers-14-02321-t005:** Future research proposals in PDAC pathology. HNF1B, hepatocyte nuclear factor-1β; PANR, PDAC-associated neural remodeling; PNI, perineural invasion; AVAC, ampulla Vater adenocarcinoma; IHC, immunohistochemistry; NRF2, nuclear factor-erythroid 2–related factor 2.

Topic	Aims
Foamy gland vs. clear cell patterns	compare morphologic and clinical aspects of foamy gland and clear cell patterns of PDAC assess the HNF1B expression in foamy gland pattern
Prognostic influence of PANR	quantitively evaluate perineural invasion (PNI) extension subgrouping PNI among involved nerves size, intra/extra-tumoral, intra/extra-pancreatic, and intraneural invasion
Large AVAC vs. PDAC tumors	compare large periampullary adenocarcinomas with PDAC regarding morphology and IHC features
Prognostic relevance in PDAC with squamous differentiation	assess prognostic value of squamous differentiation in PDAC p63/p40 expression incidence
Large duct carcinoma dedifferentiation	evaluate dedifferentiation morphology and IHC profile of PDAC with a primarily dominant large duct pattern
Comprehensive morphologic and IHC profile of PDAC	combine morphology and IHC profiles of PDAC (tubular, colloid, adenosquamous, large-duct, clear cell, foamy gland) with other proposed in the literature [17,18] to reveal PDAC subtypes include stromal compartment composition and IHC staining in the assessment
NRF2 expression in PDAC	assess independently the prognostic influence of NRF2 expression in PDAC cells correlate NRF2 expression patterns with resistance to gemcitabine (and other chemotherapeutics)
CDX2 expression in colloid carcinoma	report CDX2 staining profile among colloid carcinomas and PDAC with partial colloid differentiation

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
