# Peer review of "Current Pathology Model of Pancreatic Cancer"

_cancers, 2022, doi:10.3390/cancers14092321_

Round 1
Reviewer 1 Report
A timely review article by Dr. Szymoński and their group elaborates on the current pathological models in pancreatic cancer. Since pancreatic cancer has very less 5-year survival rates, a validated well model system is warranted and this review serves the purpose. But few things need to be addressed before it is ready for acceptance. they are as follows:
- KRAS is a major driver oncogene for pancreatic cancer which plays a significant role in cancer metabolism (PMID: 33870211 and PMID: 22541435), authors should discuss a couple of additional lines on this mentioning whether any pathological models are available to study metabolic effects in RAS-driven pancreatic cancer and what will be possible aspect on that.
- Authors should also discuss antioxidant marker NRF2's role in pancreatic cancer. recently it has been shown that NRF2 plays a non-canonical role in KRAS-mediated pancreatic cancer (PMID: 31911550 and PMID: 27477511). Authors should add their point of view on this aspect.
- Authors should also mention whether any relevant clinical trials are ongoing based on the mentioned biomarkers in this manuscript.
Reviewer 2 Report
Szymoski et al. summarize some of the latest research from the pathologist's and clinician's perspectives in the current review article "Current pathology model of pancreatic cancer." They concentrate on pathology assessment and reporting in order to make them useful in clinical as well as research settings. Finally, they discuss new diagnostic and research methodologies, as well as certain gaps in the area and future research prospects.
Overall the review article is straightforward and within the scope of MDPI-Cancers.
However, I have minor queries which should be addressed before publishing this article:
- The content summary is a bit lengthy-line 53-85. I wonder why would the authors write a special paragraph for a content summary? As a reader one would understand the contents if it is given and written in a proper fashion. The authors should reduce the content summary to a minimum. A lengthy review is not necessarily a good review.
- Throughout the review, the authors randomly start new line/s of sentences when there is no need. For example lines 455-456. Lines 453-464 can be one paragraph discussing stroma no need to start a fresh line just because the work was done by other researchers. Moreover, the fresh line starts with “Additionally”?
- Suggestion: If possible, the authors should make a nice overall diagram/flow chart about the main issues in pancreatic cancer (a picture is equal to thousand words).
Reviewer 3 Report
The authors have compiled a detailed and impressive review entitled"Current pathology model of pancreatic cancer".The topics and sub topics in the review like precursor lesions,molecular subtypes,morphological heterogeneity are well explained.The table enumerating differential pathology findings is very pertinent and useful.Immunostaining and markers of malignancy have been covered in well and convey the information easily.
The cancer stroma interactions have been discussed in depth and offer recent information.
I would like to suggest the Editor to accept the review for publication.
Round 2
Reviewer 1 Report
All concerns have been addressed, ready for acceptance.